# Emerging Application of Magnetic Nanoparticles for Diagnosis and Treatment of Cancer

**DOI:** 10.3390/polym13234146

**Published:** 2021-11-27

**Authors:** Dalal A. Alromi, Seyed Yazdan Madani, Alexander Seifalian

**Affiliations:** 1School of Pharmacy, University of Nottingham, Nottingham NG7 2RD, UK; paydaa@nottingham.ac.uk (D.A.A.); Seyed.Yazdan@nottingham.edu.my (S.Y.M.); 2School of Pharmacy, University of Nottingham Malaysia, Semenyih 43500, Malaysia; 3Nanotechnology and Regenerative Medicine Commercialisation Centre (NanoRegMed Ltd.), London BioScience Innovation Centre, 2 Royal College Street, London NW1 0NH, UK

**Keywords:** magnetic nanoparticles, cancer, synthesis, functionalization, drug delivery, magnetic hyperthermia, diagnosis, toxicity

## Abstract

Cancer is a disease that has resulted in millions of deaths worldwide. The current conventional therapies utilized for the treatment of cancer have detrimental side effects. This led scientific researchers to explore new therapeutic avenues with an improved benefit to risk profile. Researchers have found nanoparticles, particles between the 1 and 100 nm range, to be encouraging tools in the area of cancer. Magnetic nanoparticles are one of many available nanoparticles at present. Magnetic nanoparticles have increasingly been receiving a considerable amount of attention in recent years owing to their unique magnetic properties, among many others. Magnetic nanoparticles can be controlled by an external magnetic field, signifying their ability to be site specific. The most popular approaches for the synthesis of magnetic nanoparticles are co-precipitation, thermal decomposition, hydrothermal, and polyol synthesis. The functionalization of magnetic nanoparticles is essential as it significantly increases their biocompatibility. The most utilized functionalization agents are comprised of polymers. The synthesis and functionalization of magnetic nanoparticles will be further explored in this review. The biomedical applications of magnetic nanoparticles investigated in this review are drug delivery, magnetic hyperthermia, and diagnosis. The diagnosis aspect focuses on the utilization of magnetic nanoparticles as contrast agents in magnetic resonance imaging. Clinical trials and toxicology studies relating to the application of magnetic nanoparticles for the diagnosis and treatment of cancer will also be discussed in this review.

## 1. Introduction

Cancer is the biggest killer worldwide, with an estimate of 19.3 million new cancer cases and almost 10 million cancer death occurring in 2020 [1]. In the United Kingdom alone, there are over 510 deaths daily. This has devastating effects on the family, especially children, as well as all to society and economics. Deaths from cancer across the world are expected to rise, with an estimation of 12 million deaths by 2030 [2]. Although considerable progress has been done in the treatment of cancer over the last 50 years, it remains a huge health concern [3]. Consequently, the advancement of effective resources for the diagnosis, monitoring, and treatment of cancer is a continuous challenge. Some of the current therapies used for cancer treatment are radiation therapy, chemotherapy, and surgery [4]. Whilst these have been the conventional therapies used for decades, they do have their disadvantages and side effects. For example, the surgical removal of tumours is mostly limited to large, accessible, and resectable tumours. Chemotherapeutic agents only target cells that are rapidly dividing, which means they will not only kill cancer cells, but normal cells such as bone marrow cells as well [5]. Chemotherapy can result in serious side effects such as hair loss, nerve damage, nausea, and infertility [6]. Radiation therapy such as gamma rays unavoidably causes healthy tissues to deteriorate along the path of radiation. In consideration of the limitations of current treatments, it is crucial to improve cancer therapies to specifically target tumour cells and avoid healthy tissues [7]. For this reason, an enormous effort has been dedicated to researching new therapeutic approaches [3]. The application of nanomaterials for treatment of cancer is emerging as a possible viable option and has entered the phase of clinical application [8].

Nanotechnology is a field that has drawn the attention of scientific communities around the globe. The notion of nanotechnology was presented in a lecture by Nobel Laureate Richard Feynman at the California Institute of Technology, in December of 1959 [9]. In the subsequent years, various aspects of nanomaterials have been researched. This involves the engineering of these minute nanostructures, in the context of their surface chemistries, chemical composition, binding ligands, antibodies for certain activities, and lowering of toxicity levels [10]. The formation of nanomaterials is largely dependent on the cross-collaboration of several disciplines throughout science to modify and transform the atomic dimensions of materials [11].

The development of the field in the form of various nanostructures, such as quantum dots, metallic nanoparticles, fullerenes, and magnetic nanoparticles (MNPs), has attracted a considerable amount of attention from the electronic, material, and medical sciences [9]. This field has gained popularity and competitive demand owing to its subsequent social and economic impacts [12]. The relationship between nanotechnology, biology, and medicine is a fast-moving and interesting area of research. Multiple experts have indicated that the use of nanotechnology in medicine, also known as “nanomedicine”, provides numerous exciting opportunities for health care in the future and could transform the areas of tissue engineering, targeted drug delivery, and disease detection [13,14,15,16]. Early detection of diseases is very desirable in order for better health outcomes and lessening social-economic strains [17,18,19]. In particular, quick and sensitive identification of disease biomarkers will instantaneously become valuable for diagnostic screening. It will also facilitate the monitoring of disease progression and the efficiency of treatment [20,21]. Nanotechnology platforms have shown to be exceptional agents for biomedical applications (Figure 1) [22].

Nanoparticles is a term that refers to materials with one dimension at a minimum, ranging between roughly 1 and 100 nanometres (nm), normally containing a couple of hundreds to 10^5^ atoms [24]. Nanoparticles are composed of organic (e.g., polymeric) or inorganic materials which could be biodegradable. Their significance stems from the fact that the properties of nanoparticles differ from those of bulk materials formed from the same configuration. This is largely due to size effects and the role of surface phenomena with size reduction. Nanoparticles can either act at the cellular or tissular level. When acting on the cellular level, the nanoparticles can be endocytosed or phagocytosed (for example, by macrophages or dendritic cells), leading to internalization. Through this approach, the nanoparticles may transcend beyond the cytoplasmic membrane, and in some instances, beyond the nuclear membrane (for example, transfection applications) [25]. In recent times, nanoscale materials were the main focus of research, specifically in the areas of tissue engineering and regenerative medicine. A few examples include nanofibers, nanotubes, and nanoparticles, which can all be specifically adjusted to their function and purpose in tissue engineering [26]. The main advantage of utilizing nanoparticles is that they can be precisely manipulated and directed to a certain biological marker or entity and engage on a protein (3–50 nm), genetic (10–100 nm), subcellular (20–250 nm), or cellular level (10–100 nm) [27,28]. Their distinct dimensions, along with their special characteristics, have increased their attraction within this field [29]. Nanoparticles are found in nature, but can also be a consequence of human activity. They have distinctive qualities owing to their submicron size. This includes a larger surface area to volume ratio, a great value of fraction of near-surface layers and surface atoms, and the capability to display quantum effects. Their uncommon characteristics cannot be predicted based on the features of bulk materials. They are widely used in a number of technical and scientific areas, including catalysis, ecology, engineering, and healthcare [30].

MNPs are one of the most extensively researched nanomaterials, owing to their potential uses in various areas of research [31]. MNPs have already been used for the detection of cancer by localising the sentinel node, as well as molecular imaging [32]. MNPs are extensively being researched for usage in various industrial and scientific areas, varying from mass data storage to catalysis. The concept of using MNPs to target tumour cells within the human body for the treatment of cancer was first suggested late in the 1970s [33]. MNPs can be used to heat up the tumour and kill it, which is due to tumours cells being more sensitive to temperature increase than healthy ones. By getting the MNPs into the tumours, then releasing energy as heat when subjected to an alternating magnetic field, this will damage the cancer cells [34]. Those magnetic nanomaterials are categorised into five main types: ferromagnetic (such as cobalt, nickel, and iron), paramagnetic (such as magnesium, lithium, tantalum, and gadolinium), diamagnetic (such as silver, copper, gold, and the majority of known elements), antiferromagnetic (such as CoO, MnO, CuCl_2_, and NiO), and ferrimagnetic (such as maghemite γ-Fe_2_O_3_ and magnetite Fe_3_O_4_) [24]. The chemical and physical characteristics of MNPs are significantly reliant on their size, shape, crystalline structures, and chemical components. Additionally, MNPs have special magnetic properties such as low Curie temperature, superparamagnetism, and great magnetic susceptibility [35]. Magnetic susceptibility is the ratio of magnetization to an applied field that demonstrates how strongly a nanoparticle is either attracted to or repelled by a magnetic field [36].

In regard to MNPs, the fundamental idea was to affix conventional anticancer drugs to small magnetic spheres externally, prior to administering them into the human body. Once injected into the blood stream, strong external magnetic fields would gather the drug loaded nanoparticles within the tumour tissue. It is expected that with this method, the drug payload will be considerably decreased. Therefore, the undesirable side effects linked to the systematic distribution of chemotherapeutics, including hair loss, nausea, and a weakened immune system, would be prevented. Despite the fact that it is still not fully in clinical use, nanomedicine has made great strides from these original concepts and is progressing at a phenomenal speed [33].

MNPs ranging from 10 to 100 nm are favourable for use in-vivo, considering they do not present with rapid renal clearance, as nanoparticles smaller than 10 nm do, and do not become internalised by the reticuloendothelial system, as nanoparticles over 200 nm do [37]. All MNPs utilized in-vivo thus far are comprised of the iron oxides maghemite (γ-Fe_2_O_3_) and magnetite (Fe_3_O_4_), owing to their recognized pathways of metabolism and their low levels of toxicity. Both oxides’ crystal structures are based upon a cubic dense packing of oxygen atoms, only varying in their distribution of iron ions inside the crystal lattice [38]. Iron oxide MNPs are the most favourable nanomaterials in medical sciences as a result of their characteristics of biocompatibility, stability in aqueous solutions, low toxicity, and brilliant physiochemical properties such as superparamagnetism [9]. The widespread use of iron oxide MNPs is also attributed to their ability to manipulate particle motion, cause energy dissipation, and provide imaging contrast upon exposure to an external magnetic field [39]. The magnetic response of iron oxide is stable due to its low sensitivity to oxidation [40]. Moreover, iron oxide MNPs have an advantage over alternative metal nanoparticles because of their size control, specific interaction and dispersion, and avoidance of aggregation by coating and penetration of cell and tissue barriers [9]. Overall, MNPs acquired greater attention because of their unique behavioural, structural (Figure 2), and diversified applicable qualities. For instance, their distinct magnetic properties and adjustable size, functionalizable surface with various molecules, biocompatibility with different cell types, high chemical stability with increased surface area, inductive magnetic moment, high magnetic susceptibility and superparamagnetism [22]. Nanoparticles that are superparamagnetic and biocompatible are immediately injected into tumour tissue, where they can be controlled by an external magnetic field to generate heat as a consequence of the Brownian and Néel relaxation processes [38].

In the interest of further expanding the opportunities these MNPs can offer, researchers have attempted to adjust their magnetic properties through modifying their size, shape, and morphology. It should be noted that in recent times MNPs with an internal cavity have been invented, creating hollow structures. From the perspective of magnetism, this “hollow” structure is particularly fascinating because the existence of both inner and outer surfaces leads to MNPs with enhanced total surface areas. This results in increased surface disorder and thus higher surface anisotropy and exchange bias. Recently, various types of hollow MNPs with improved qualities have been announced in the literature. A majority of them are based on magnetic oxides and ferrites. Hollow MNPs with adjustable shell thickness and composition are excellent constituents for new enhanced materials that can be used for a lot of possible applications. In theory, hollow structures allow the encapsulation of various contents inside the MNPs for different applications. For instance, the hollow MNPs may be encapsulated with anticancer drugs and used for drug delivery applications [41]. In comparison to normal nanoparticles, hollow nanoparticles take advantage of the greater surface area and pore volume, as well as the additional paramagnetic centres, meaning that they could yield an enhanced longitudinal relaxation performance [42]. Essentially, metal nanoparticles exemplify an important gateway for the future of medicine [23]. Hence, MNPs will be the primary focus of this review [30].

A number of approaches were suggested for the synthesis of MNPs, i.e., reverse micelles synthesis, co-precipitation, hydrothermal synthesis, and thermal reduction or decomposition [9]. Thus far, methods based upon alternative mechanisms, such as the nanoscale Kirkendall effect, were highly developed to synthesise several hollow nanoparticles, for instance hollow iron oxide nanoparticles [42]. Following the synthesis of MNPs, various surfactants and polymer coatings were utilized, for example dextran and polyethylene glycol (PEG) [9]. The functionalization of MNPs is desirable, as it will improve their biocompatibility, protect their magnetic core from oxidation, and prevent the nanoparticles from agglomerating.

In the last 20 years, MNPs have been applied in various biomedical applications such as drug delivery, cancer therapy, and magnetic resonance imaging (MRI) [43]. In addition, hollow MNPs have several biomedical applications of their own. This involves the simultaneous provision of diagnosis and therapy, since the large cavity found within the hollow nanostructure could be utilized to contain different biomolecules and drugs that are released in a regulated manner. Furthermore, the surface of hollow MNPs could be functionalised with targeting agents [44].

Although there have been many developments in the area of nanomedicine in recent years, there is still a lot of drawbacks. For instance, the long-term toxicity of several nanoparticles is not widely known owing to the newness of this field of research [23]. Additionally, despite it not entirely being in clinical use yet, nanomedicine has made great progress and is advancing at extraordinary speed [45].

The aim of this research was to investigate the application of magnetic nanoparticles for the diagnosis and treatment of cancer. In this review, various aspects of magnetic nanoparticles will be explored to aid in this investigation. The synthesis techniques and functionalization approaches of magnetic nanoparticles will be discussed. Clinical trials and toxicology studies of magnetic nanoparticles will also be reviewed.

## 2. Method

This study is a systematic review with an aim of evaluating the use of MNPs in the treatment and diagnosis of cancer. Subsequent to the research, a reiterative approach apprehended a search strategy and study selection, along with inclusion and exclusion criteria, resulting in effective data collection.

### 2.1. Search Strategy

Articles published up until July 2021 were researched via digital databases worldwide containing PubMed and Science Direct. The keywords involved “magnetic”, “nanoparticle”, “synthesis”, “toxicity”, “functionalization”, “cancer”, “biomedical”, “applications”, “drug”, “delivery”, “MRI”, “hyperthermia”, “treatment”, and “diagnosis”.

### 2.2. Study Selection

I reviewed papers in order to assess their eligibility. I followed a protocol in order to do so. I initially reviewed papers based on their abstracts. If I considered the abstract to be relevant, I then proceeded to review the paper in full. Finally, I selected the information I deemed appropriate to include in my review.

### 2.3. Inclusion Criteria

The articles that complied with the following requirements were included in the study: published articles, articles written in English, articles discovered by searching for keywords, and articles containing enough information about the synthesis, functionalization, biomedical applications, clinical trials, or toxicity of magnetic nanoparticles. Review papers were also included in this paper.

### 2.4. Exclusion Criteria

The articles that complied with the following requirements were excluded from the study: articles not written in English, articles that do not include the keywords mentioned, and articles that do not contain enough information about the synthesis, functionalization, biomedical applications, clinical trials, or toxicity of magnetic nanoparticles.

### 2.5. Data Collection

Data was collected based on the following factors: period of publication, magnetic nanoparticles, synthesis methods, functionalization techniques, toxicology, and biomedical applications. Data collection was conducted by one researcher. The researcher then reviewed the data collected. Data was collected relating to their various biomedical applications, by assessing papers reviewing the application of MNPs.

### 2.6. Synthesis of Magnetic Nanoparticles

The synthesis of MNPs is a multistep process that requires great attention in order to obtain its desired results [46]. There are various approaches to synthesizing MNPs. MNPs can either be produced via “top-down” or “bottom-up” techniques [47]. The “top-down” approach comprises of high energy ball milling of magnetic samples, until desirable nanoscale sizes are attained. The benefit of this approach is its ability to attain a large number of particles in an individual batch. The drawback is its lack of control over the particle’s shape and size, which is an important aspect to biomedical applications [48]. The “bottom-up” approach could begin with either a ferrous (Fe^2+^) or Ferric (Fe^3+^) ion salt. The salt then undergoes a separate chemical procedure to nucleate and stimulate seeded growth in order to grow particles to their required hydrodynamic diameters [49]. Various “bottom-up” methods are stated in the literature. The most reported approaches are co-precipitation, hydrothermal synthesis, thermal decomposition, and polyol synthesis (Table 1) [22]. Different approaches include microemulsion, preparation with micelles, solvothermal, sol-gel, flow injection technique, sonochemical, microwave-assisted, physical vapor deposition, chemical vapor deposition, electrodeposition, laser pyrolysis, combustion, and carbon ARC [50]. In this review, we will focus on co-precipitation and hydrothermal synthesis, as they are the most commonly used synthesis techniques.

### 2.7. Co-Precipitation

The co-precipitation method is commonly used for multiple biomedical applications owing to its simple preparation procedure and easy application. Additionally, fewer toxic precursors are required. Co-precipitation is typically performed in an aqueous medium, with salt solutions and a base, to form insoluble solid particles. This is achievable with or without the addition of a precipitating agent [46]. Following this procedure, the synthesis of MNPs can be carried out at room temperature or a high temperature, which will result in a high yield, as well as different shapes and sizes [51]. The shapes and sizes of the MNPs are dependent on multiple components. For example, the pH value of the solution, reaction temperature, type of salt used, etc. [52]. The pH must range from 8 to 14 for effective precipitation [51], although a rise in both pH value and ionic strength in the aqueous medium will reduce in the size of MNPs [46] as both those components modify the nanoparticle’s electrostatic surface charge and chemical structure surface [22].

Usually, co-precipitation begins with a ferrous and ferric salt ratio of 2:1 and a basic condition, either at room temperature or a temperature ranging between 80 °C and 85 °C. The basic condition is reached by the use of various bases, for example NaOH. Once the reaction is complete, precipitation is observed at the base of the reactor. The following process of washing, drying, sintering, and grinding produces the MNPs [22].

The main concern with this approach is the insufficient control it has over size distribution. Narrow size distribution is vital for complete utilization of magnetization in future applications, given that magnetization is heavily dependent on the size of nanoparticles. The addition of organic stabilizing agents may offer partial control over size distribution, and the most widely used stabilizing agent for MNPs is oleic acid [33].

### 2.8. Hydrothermal

Hydrothermal synthesis, also known as solvothermal synthesis, is the most commonly known wet chemical method used to create inorganic nanoparticles, specifically metals and oxides. The hydrothermal process typically includes wet chemical approaches for crystallization in a sealed container. In the container, an aqueous solution is maintained at a high temperature, ranging between 130 °C to 250 °C, as well as a high- pressure, ranging between 0.3 MPa to 4 MPa. Hydrothermal synthesis normally generates nanoparticles with larger diameters. In one study, a diameter of 27 nm was formed for Fe_3_O_4_ nanoparticles with the use of a surfactant like sodium bissulfosuccinate. Alternately, when this technique was utilized for 6 h at 140 °C, a diameter of 40 nm was produced for Fe_3_O_4_ powder [46]. This hydrothermal approach allows for the adjustment of nanomaterials from several nanometres to a hundred nanometres [22]. Typically, the reaction temperature, concentration of precursors, and total duration of reaction controls the size and distribution of the synthesized nanoparticle. The hydrothermal method provides numerous benefits, such as exceptional crystallization and simple morphology control of the product. Several researchers have proven that MNPs of diverse shapes, for instance nanowires and nanospheres, can be produced through the hydrothermal technique [53]. The hydrothermal method is deemed an environmentally friendly and multi-faceted approach, since organic solvents are not used, and nanoparticles do not require treatment following synthesis [54]. The most significant disadvantage of this technique is its failure to acquire nanoparticles smaller than 10 nm. Another drawback is its delayed reaction kinetics at elevated temperatures [46].

**Table 1 polymers-13-04146-t001:** Advantages and disadvantages of the most popular methods used to synthesize MNPs.

Synthesis Method	Advantages	Disadvantages	Year of Study
Co-precipitation [55,56]	Fast reaction Easily scale up the production.	Surface oxidation Poor reproducibility	2013, 2016
Hydrothermal [55,57,58,59]	Magnetic controllability Excellent control of size, shape, and dispersion	Adsorption of capping agents Prolonged synthesis duration	2008, 2013, 2017, 2019
Thermal decomposition [55,57,59,60]	Great reproducibility Excellent size distribution	Toxicity Soluble in organic solvents	2008, 2013, 2019
Polyol [54,59,61,62,63]	Biocompatibility Cost-effective industrial application	Unstable oxidation Complex synthesis of small particles	2009, 2017, 2018, 2019

### 2.9. Functionalization of Magnetic Nanoparticles

Functionalization is a fundamental component in the synthesis of MNPs. It allows for the utilization of MNPs in biomedical applications (Table 2). In the absence of functionalization, MNPs would be unstable in strongly acidic environments and prone to leaching, decreasing their lifespan and limiting their reusability [64]. The main objective of functionalizing MNPs is to improve solubility and biocompatibility, enhance surface catalytic activity, avoid agglomeration, and enhance physiochemical and mechanical qualities [65]. Additionally, the functionalization of MNPs is vital in order to avoid oxidation. This is particularly an issue for pure metals such as iron, nickel, cobalt, as well as their alloys, due to their sensitivity to air [33]. For example, the oxidation of magnetite (Fe_3_O_4_) into maghemite (γFe_2_O_3_) could give rise to alterations in their magnetic properties [57].

Functionalization could be achieved either in-situ or post-synthesis. The in-situ approach refers to the concurrent synthesis and functionalization of MNPs, whereas the post-synthesis technique implies that functionalization occurs after synthesis. The three approaches to functionalization are encapsulation, ligand exchange, and ligand addition [66]. Encapsulation is the most commonly used method, owing to the variety of coating agents available to be used. Organic materials, such as polymers and surfactants, and inorganic materials, such as carbon, silica, metal, and metal oxides, may be employed for encapsulation. The utilization of polymers for functionalization is the most widely used approach for biomedical applications, specifically in nanomedicine. The most commonly used coating agent in the context of inorganic encapsulation is silica. The Stöber method, aerosol pyrolysis, microemulsion, and sodium silicate-based approaches are the most frequently used silica encapsulation techniques [65]. Functionalized MNPs are reasonably biocompatible, possess hydrophilic attributes, and have been utilized for multiple biomedical applications such as drug delivery [65,67]. Alternative inorganic coating agents such as carbon, metals, and metal oxides have limited biomedical applications [65]. The focus of this section will be on polymer, silica, gold, and carbon coating agents, polymer and silica coatings being the most popular functionalization approaches.

**Table 2 polymers-13-04146-t002:** Several functionalization agents used on MNPs for drug delivery and diagnosis. The check mark represents the biomedical application it is used for.

Functionalization Agent	Drug Delivery	Diagnosis	Year of Study
Polyethylene glycol (PEG) [65,68,69]	✓	✓	2004, 2016, 2018
Polyethyleneimine (PEI) [65,70]	✓		2018
Polyvinylpyrrolidone (PVP) [65]	✓	✓	2018
Polyvinyl alcohol (PVA) [54,65,71]	✓	✓	2016, 2018, 2019
Dextran [54,65,72]	✓	✓	2018, 2019
Chitosan [54,65,73]	✓	✓	2018, 2019
Silica [53,54,65,67,74]	✓		2012, 2014, 2015, 2018, 2019
Carbon [54,75]	✓		2017, 2019

### 2.10. Polymer Coating Agents

The most frequently used polymers for the functionalization of MNPs are chitosan, alginate, dextran, polysaccharide, polyacid polyetherimide, polyethyleneimine (PEI), polydopamine (PDA), polyvinylpyrrolidone (PVP), polyamidoamine (PAMAM), polyethylene glycol (PEG), and polyvinyl alcohol (PVA) [65]. PEG is the best-known coating agent [76]. It enhances biocompatibility and improves the cellular uptake of MNPs. It is worth noting that the polymer layer’s thickness is important. If the layer is not thick enough, it might not be able to protect the MNPs from oxidation. Atom transfer radical polymerization is a great approach to control the polymer layer’s thickness [43].

Amphiphilic polymers have been utilized to provide nanoparticles with aqueous dispersibility qualities and to couple biomolecules onto the surface of MNPs. Normally, carbonyl groups are extensively used to enhance water solubility. The utilization of amphiphilic polymers is direct and quick, given that it merely depends on hydrophobic interactions [77].

Polysaccharides such as dextran and alginate have effectively been applied as coating agents. Chitosan increases particle adhesion to cells and enhances biocompatibility. However, polysaccharides demonstrate poor mechanical resistance and the microemulsion method cannot be implemented for in-situ particle coatings, since polysaccharides have difficulty solubilizing in organic solvents. Moreover, to enhance the stability of MNPs in organics, non-polymeric stabilizers, for instance fatty acids, have favourably been utilized [43]. Additionally, MNPs may be encapsulated within liposomes to allow for improved uptake in cells for gene and drug delivery [78].

Studies have illustrated that utilizing two polymer layers or copolymers could enhance biocompatibility and/or the effectiveness of drug delivery [43]. For example, MNPs with β-cyclodextrin and pluronic polymer as coating agents have shown enhanced MRI efficacy and increased hyperthermic effects, in comparison to MNPs coated with β-cyclodextrin alone [79]. Moreover, new characteristics can be obtained by connecting two diverse polymers. For example, PEG-PEI copolymers demonstrate stealth behaviour and DNA binding affinity, owing to the properties of PEG and PEI, respectively [43].

### 2.11. Silica Coating Agents

Silica coating is an alternative functionalization approach to enhancing the biocompatibility and stability of MNPs [80]. It is achieved via the utilization of silanol groups that readily react with alcohols and silane coupling agents. Aminopropyl trimethoxysilane, (3-mercaptopropyl) triethoxysilane, and (3-Aminopropyl) triethoxysilane are the most commonly utilized agents for the introduction of functional groups for additional derivatization [43].

The exterior of silica coatings could be functionalized by utilizing the large concentration of hydroxyl surface groups to offer intrinsic hydrophilicity and enable certain biomolecules to bind to the surface via covalent bonds [25]. In addition, the interior of silica coatings could be utilized to encapsulate certain drugs, whilst preventing the undesirable physical adsorption that occurs with larger molecules. As a result, MNPs with silica coatings can be employed for therapeutic and diagnostic purposes [25,43].

Silica coatings are achieved via the hydrolysis and condensation of tetraethoxy silane, also known as the Stöber method, through sol-gel or microemulsion methods. Based on the synthetic parameters, particularly pH, either singular silica coated MNPs or silica nanoparticles encapsulated with multiple MNPs can be acquired. The silica layer’s thickness can be manipulated up to 2 nm [81].

Tumour inhibition has been shown in vitro by mesoporous silica MNPs embedded with RNA molecules. Silica coatings also have a defensive role, inhibiting interactions between MNPs and molecules attached onto their outer surface [43]. Moreover, silica coated nanoparticles are negatively charged at the blood’s pH, which induces electrostatic repulsion and prevents aggregation. Silica is also heat resistant, has a large surface area, and optimal mechanical strength [25]. Furthermore, the ligand- exchange method can be utilized to produce PEG-silanes, in order to create hydrophobic MNPs coated with oleic acid that readily disperse in aqueous mediums. However, silica is unstable in basic conditions and fine control over silica deposition is hard to accomplish [43].

### 2.12. Gold Coating Agents

Gold coating agents are utilized because of their ability to shield metal cores, such as iron, from oxidation and enable additional functionalization. Moreover, gold coatings can be employed in cancer therapy owing to their capability of converting energy into heat, through alternating radio waves, magnetic fields, or near-infrared light [43]. This focused heat production is effective against tumours resistant to chemotherapy, or for enhanced therapeutic effectiveness when amalgamated with conventional anticancer drugs or radiotherapy [82].

Gold deposition is normally performed via the reduction of Au(III) on the external surface of MNPs, such as iron oxide nanoparticles. Alternate authors used sonochemical techniques to create iron and gold core-shell nanoparticles with a limited size distribution [82]. The first report of gold coated MNPs was in 2001, when gold coated iron nanoparticles, with a diameter of 18–80 nm, were created through the reverse micelle mechanism. To prevent the nanoparticles from aggregating, 1-dodecanethiol (C12H25SH) was attached onto the surface of the gold coating, via a self-assembly process. Gold coated MNPs can be functionalized to bind biomolecules through employing thiol linkers at the opposite end of the molecules with a functional group, for instance, amine [83].

### 2.13. Carbon Coating Agents

Carbon and its derivatives have been used for coating the MNPs. This is due to biocompatibility, chemical stability, and substantial thermal capacity. Carbon coated MNPs demonstrate a larger magnetic moment in comparison to their equivalent oxides, since they are normally found in their metallic form. However, synthesis techniques that produce carbon coated MNPs frequently result in agglomeration. Furthermore, carbon coatings are usually heterogenous and precise control over the thickness of the coating is yet to be accomplished [43].

However, recently, functionalised graphene oxide (FGO) has been used to coat the MNPs, due to functionalisation of GO and bonding to the MNPs, which eliminates the agglomeration of the nanoparticles [84].

### 2.14. Applications of Magnetic Nanoparticles for Diagnostic and Treatment of Cancer

MNPs are an optimal choice for drug delivery owing to their low toxicity levels, great targeting efficiency, and large surface area to volume ratio [85]. Additionally, MNPs can be utilized in magnetic hyperthermia to destroy cancer cells and decrease tumour volume via a targeted approach [86]. Moreover, the magnetic qualities of MNPs can be employed for significant imaging modalities, such as MRI [22]. The imaging qualities of MNPs make them an ideal option for the concurrent provision of diagnosis and therapy, also known as theranostics [86]. MNPs offer a greater theranostic capacity in comparison to liposomes or alternative polymer-based nanoparticles, due to their magnetic properties [87]. MNPs are beneficial in theranostics as a result of their ability to concurrently be directed, visualized, and heated by external magnetic fields [43]. This section will discuss the application of MNPs in drug delivery, magnetic hyperthermia, and diagnosis (MNPs applied as MRI contrast agents), since they are the most widespread applications of MNPs at present (Figure 3).

### 2.15. Application of Magnetic Nanoparticles in Drug Delivery

The great extent of non-specificity associated with the utilization of drugs is a significant disadvantage. Following oral or intravenous administration, the systemic distribution of drugs leads to the development of adverse effects, and less of the active drug reaches the target site. Thus, larger doses are required when administering conventional anti-cancer drugs in order to achieve an appropriate local concentration at the site of action. This is a major problem, particularly for anti-cancer drugs that demonstrate severe adverse effects, such as cardiomyopathy, neurotoxicity, hair loss, and bone marrow suppression [88].

Multiple strategies relating to the delivery of drugs to tumour regions were considered. In 1960, it was suggested that MNPs could be transported via the circulatory system to a specific site within the body, with the guidance of a magnetic field. After the 1970s, advancements in the application of MNPs for the delivery of chemotherapeutic agents was observed [25].

Nanoparticles utilized in drug delivery can simultaneously enhance drug stability, and overcome the issues associated with the administration of conventional anti-cancer therapies. For example, superparamagnetic iron oxide nanoparticles (SPIONs) functionalized with PEG and conjugated with doxorubicin (SPIO-PEG-D) were developed for chemotherapy. By conjugating doxorubicin onto the surface of SPIONs with PEG, the half-life of doxorubicin was extended. An in-vitro experiment demonstrated that SPIO-PEG-D results in reduced DNA expression and increased cell apoptosis for HT-29 cancer cells. Additionally, in-vivo experiments illustrated the reduction of cardiotoxic and hepatotoxic side effects due to the combination of this drug delivery system and an external magnetic field [89].

MNPs achieve controlled and specific drug release by attaching to drug molecules via a cleavable linker, or a polymeric shell created with the ability of releasing drugs. Those MNPs can be guided to a specific site by the application of an external magnetic field, and the drugs can be delivered as a result of either enzymatic cleavage or modifications in the physiological environment, such as temperature or pH [43]. For example, nanoparticles were synthesized with methotrexate and dendrimer- doxorubicin attached onto their surface via amide or cleavable hydrazone bonds [90]. Therefore, when the nanoparticles are taken up into the cells, the bonds cleave owing to the presence of lysozymes. Through passive targeting (Figure 4), the nanoparticles improved the delivery of doxorubicin to the tumour site, and drug release only took place at lysosomial pH [43].

Targeted delivery methods are segregated into two categories, passive and active. Passive methods do not require external forces, whereas active methods require an external energy source to guide the nanoparticles to their site of action [91]. The effectiveness of the targeted delivery method is dependent on a variety of variables. For instance, the injection procedure, MNP concentration, hydrodynamic conditions, the qualities of the magnetic field, MNPs (includes drug-particle binding), and the target area’s location and depth [24]. One of the major issues with the application of MNPs is the depth in which the magnetic field is able to penetrate. The magnetic field can penetrate the body up to 2 cm from the skin without difficulty (Figure 5), however penetrating the body beyond 2 cm is difficult, since the magnetic field reduces with distance [30]. Nevertheless, internal magnets can be situated in the tumour’s proximity through minimally invasive surgery to evade the restrictions of external magnetic fields. Multiple studies have shown simulated interactions between MNPs and magnetic implants that enabled drug delivery. With respect to all biomedical applications, there are issues relating to the extrapolation of data from animal models to humans. There are multiple physiological measures that complicate this, such as variations in weight and cardiac output. However, a great deal of interest in this area persists, considering the accessibility nanoparticles provide to specific tumour sites, in comparison to conventional surgery [25].

In recent years, fluorescent magnetic nanoparticles (fluro-MNPs) were presented to be great vectors for drug delivery. Their size and concentration in tumours provide precise mapping of lesions and very high resolution, signifying their importance in biomedical applications. This combination of magnetic and fluorescent qualities in nanoparticles is significant for multi-functional contrast agents in medical bio-imaging [92].

The production of fluro-MNPs in an aqueous solution was performed with the addition of fluorescein iso-thiocyanate (FITC) during fabrication. Fluorescence spectrophotometry and confocal microscopy detected an intensity of fluorescence in fluro-MNPs, in comparison to MNPs developed without FITC. To be appropriate for drug delivery, the fluro-MNPs were functionalized by five layers of PEG/CMC. The future use of fluro-MNPs for medical fluorescence bio-imaging and anti-cancer drug delivery is promising, given that fluro-MNPs have demonstrated that their surface is appropriate for functionalization [92].

### 2.16. Application of Magnetic Nanoparticles in Magnetic Hyperthermia

Hyperthermia is the gradual increase in temperature to 40–43 °C. It leads to the destruction of cancer cells and improves the outcomes of chemotherapy and radiation. The drawback of this approach is its inability to heat cancer cells locally. However, this issue can be bypassed by the intravenous administration of MNPs targeted towards specific sites, and the use of an external magnetic field to generate heat at a local level. This targeted technique could enhance the safety and efficiency of hyperthermia, since it does not cause damage to healthy neighbouring tissues [30].

Magnetic hyperthermia is a non-invasive approach to cancer therapy [22]. This approach offers an alternative for certain cancers that may be difficult to surgically remove and for those that reside proximally to vital organs. Since magnetic hyperthermia resolves the issue of nonselective ionizing radiation associated with conventional radiotherapy, treatment options may broaden beyond certain tumours [93]. Magnetite and maghemite are the most utilized nanoparticle materials for magnetic hyperthermia [30]. Magnetite is one of the most significant magnetic materials, due to its high saturation magnetization and its ability to easily functionalize biomolecules [94]. Magnetic hyperthermia is dependent on a rise in temperature in the tumour site, either to 41–47 °C for the induction of apoptosis, or to 50 °C for the induction of necrosis [22]. Temperatures up to 42 °C may be referred to as mild hyperthermia and higher temperatures may be referred to as extreme hyperthermia. The heat from higher temperatures may result in alterations to the permeability of the cell membrane, immune system stimulation, the denaturation of proteins, cytoskeletal damage, and impairment of specific DNA repair processes [95]. By utilizing an environment similar to the tumour microenvironment, it was shown that magnetic hyperthermia requires a target temperature about 6 °C lower than exogenous hyperthermia in order to achieve equivalent cell death effects. It also presents with cytotoxic effects of greater significance. Cancer cells are deemed to be more susceptible to heat in comparison to healthy cells, as a result of their increased rate of metabolism. On a tissular level, a tumour’s ability to disperse heat is reduced due to its disordered vascular system. Increased temperatures also enhance cell sensitivity to alternative treatments such as chemotherapy and radiation [96]. Moreover, the increase in cancer cell sensitivity can be attributed to the improved anti-tumour immune response caused by hyperthermia due to its ability to enhance the presentation of tumour antigens, trafficking of leukocytes throughout the endothelium, and NK cell and dendritic cell activation [93]. The degree of sensitivity is dependent on the time taken between heating and chemotherapy, drug nature and concentration, tumour type, and tumour temperature [97]. Hyperthermia treatment induces certain biological effects within cancer cells, such as enhanced lysosomal permeability, which in turn culminates in an increase in oxidative stress due to the production of reactive oxygen species. The reduction of tumour cell viability through amplified cathepsin D activity within the cytoplasm is another consequence of increased lysosomal permeability. Furthermore, the rotation of SPIONs induced by dynamic magnetic fields may disrupt lipid membrane stability, thereby influencing lysosomal permeability and resulting in the activation of apoptosis [93].

Brownian and Néel relaxation explain the rise in temperature found in particles exposed to an external magnetic field, during the process of magnetic hyperthermia [98]. Neel relaxation converts energy from an external magnetic field into heat generation and Brownian relaxation results in the rotation of MNPs leading to cell damage [99]. The alternating magnetic field permeates through tissue, allowing tumours to be treated in various positions within the body. The effectiveness of heating is dependent on the particle’s response to an external magnetic field, the frequency and amplitude of the magnetic field, and particle size. The particle’s resistance against the magnetic field, generates heat within the particles. The particle’s transfer of energy, magnetic to thermal, can be measured as a specific absorption rate (SAR) [22]. The injected dose administered to the patient reduces as the SAR increases [96].

The value of SAR is dependent on the frequency and amplitude of the magnetic field, the shape and size of the particles, as well as their magnetic qualities. A study stated that graphene oxide adjustable magnetic nanorods applied in mice models were efficacious for hyperthermia. The 350 nm nanorods had the greatest SAR value of 1045 W g^−1^ at 0.2 mg mL^−1^ of iron concentration, amongst the three various nanorods that were utilized (250, 350, and 460 nm). The 350 nm nanorods displayed an adequate decrease in tumour volume in the mice model and demonstrated favourable biocompatibility in the MTT assay [22]. Intrinsic loss parameter (ILP) is an additional concept to consider. A number of researchers began reporting ILP instead of SAR because it eliminates the influence of field and frequency in the calculation. ILP is only deemed constant in conditions of low field strength and low frequency measurements. Consequently, utilizing ILP as a comparative tool between studies is not advised. In favour of successfully comparing SAR performances of samples, the field, frequency, media, and measured concentration must all be kept constant [100].

In recent times, in-vivo experiments were implemented to research the effects of magnetic hyperthermia on the size of tumours. A group of researchers utilized carboxydextran coated superparamagnetic iron oxide nanoparticles for magnetic hyperthermia. BALB/c nu/nu athymic mice injected with non-small cell lung cancer cell line A549 cells were exposed to an alternating magnetic field for 20 min. The tumour size was reported to be significantly reduced. Correspondingly, researchers injected murine pancreatic carcinoma cell line Pan02 cells into mice to synthesize pancreatic cancer tissue. Iron oxide nanoparticles loaded onto monocyte/macrophage-like cells were injected into the mice following tumour growth. After three days, the MNPs found in cancerous tissue generated heat upon exposure to an alternating magnetic field. After a period of time, a substantial decline in tumour size was recorded [101]. Additionally, Sadhuka et al. have recently demonstrated that hyperthermia aids in the eradication of cancer stem cell populations, which are a significant cause of metastasis and the recurrence of cancer. This could be due to the production of reactive oxygen species [102].

Despite its initial clinical application, hyperthermia has presented various drawbacks that should be addressed for its implementation to become widespread. For hyperthermia to compete with conventional cancer therapy, synergetic effects must be demonstrated via its application with radiotherapy and chemotherapy, however its limited effectiveness hinders this process. Uninhibited heat dispersion results in reduced effectiveness, which is mainly due to the absence of powerful devices capable of controlling and surveilling local temperatures. This may be worsened by the distance between the heat source and the target tumour cells, the circulatory system producing thermal dissipation, and the inhomogeneity of the tissues. Since hyperthermia is more complicated than standard therapies, efforts have been made for the development of new specific equipment for hyperthermia treatment. In addition, hyperthermia may cause local side-effects as the heat source is not completely adjacent to the cancer cells being treated. Consequently, undesirable toxicity may occur within healthy cells or tissues due to the increase in temperature. Therefore, it is essential that thermal biological studies are undertaken to gain a further understanding. The thermotolerance and the effect of hyperthermia on cells on a molecular level, including the sensitives of various cancer types to temperature, should also be assessed [93].

These issues may be tackled through the utilization of a contactless stimulus that localizes heat induction and via the enhancement of the real-time control of temperature. Magnetic resonance thermometry (MRT) is a successful non-invasive MRI-based technique that includes the measurement of temperature distributions in 3D, potentially replacing the current utilization of invasive thermal probes. Forty years ago, the combined use of MRI with thermometry was suggested. MRT techniques have since enhanced their accuracy for their application in vivo and pre-clinical developmental stages. Novel non-invasive targeted sources of heat have been investigated for hyperthermia to further increase its effectiveness and to evade undesired side-effects. Although a single equipment enabling the simultaneous performance of both hyperthermia and in-vivo location would be advantageous, the varying magnetic field requirements between MRI and hyperthermia prohibit guidance in real-time [93].

### 2.17. Application of Magnetic Nanoparticles in Diagnosis

One of the challenges faced in medicine is the attainment of early diagnosis for different diseases like cancer. Different devices such as MRI, CT scan, and X-rays have been used over the years. High soft tissue contrast, spatial resolution, and the penetration depth of MRI have given this non-invasive instrument the advantage of becoming widely used for non-invasive clinical diagnosis [103]. MRI utilizes the combination of a large magnetic field, radiofrequencies, and a computer to produce images of tissues. Despite the widespread use of MRI, researchers are investigating different ways of improving the quality of MRI so that tissues can be detected with better resolution. To gain a better understanding, the MRI mechanism needs to be briefly discussed. In general, the interaction of three magnetic fields allows the MRI to image the tissues. The three fields are statistic field or the main magnet, gradient coils, and a dynamic radio frequency (RF) field. The nuclei in tissues move randomly and do not produce any magnetic effect in the absence of the magnetic field. However, placing the subject within the MRI magnetic field results in the parallel alignment of the protons of the tissues with the main magnet [104]. Following the alignment of protons with the magnetic field, the RF pulse is then applied to the tissue. This results in the direction of proton spins from the core magnet to be altered. The duration and strength of the RF pulse can influence the change of the proton’s direction. The RF pulse could tilt the net magnetization by 90 or 180 degrees. The transverse plane is referred to as a situation whereby the proton’s direction alters by 90 degrees. Following the termination of the RF pulse, the proton spins return to their original longitudinal direction. This results in the release of electromagnetic energy, which can be detected by the MRI. This is known as the relaxation process. The rate of energy release differs in different organs, allowing the MRI to be used as a clinical diagnosis instrument [105]. The time it takes for the proton’s spins to change and return to its original direction is indicated by T1 [32]. T1 varies in different tissues, and this is dependent on factors such as the size and motion of materials. For example, water has a high molecular motion in comparison to lipids, therefore it has a greater T1 value [106]. As previously mentioned, following the RF pulses, the proton spins to a new direction. However, this position is temporary and unstable. The time taken for the deterioration of this new direction is determined by T2 [107]. Different clinical protocols can be employed to quantify the T1 or T2 relaxation time [108].

To simplify, researchers have classified the MRI sequences into T1 and T2 divisions. These divisions were established based on the influence of different sequences on tissues. T1 is defined as longitudinal, or the time constant which determines the rate at which excited protons return to equilibrium. T2 is defined as the time constant that regulates the rate at which excited protons reach equilibrium [43]. Upon screening an organ in the MRI, the T1 scan illustrates fats as high intensity pixels and T2 displays all fat and water as high intensity pixels. The combination of both images is used for showing tissues [109]. As discussed earlier, improving the quality of images obtained from MRI is one of the main goals of various research groups. The utilization of contrast agents is one way of improving the quality of MRI images [110]. The contrast agent helps by increasing the difference between normal and abnormal tissues. The first contrast agent was ferric chloride, which was used for the MRI of the gastrointestinal tract in 1984 [111]. Researchers investigated the application of different contrast agents such as gadolinium, manganese, and dysprosium. However, the slow relaxation and the presence of seven unpaired electrons in gadolinium have made this material the most popular option for contrast agents [112]. Contrast agent application is achieved via shortening the T1 and T2 relaxation time. Contrast agents that alter the T1 relaxation time are known as positive contrast agents, for instance gadolinium. The shortening of the T1 relaxation time results in organs becoming bright in the T1 scan of MRI [105]. On the contrary, contrast agents that shorten the T2 relaxation time are called negative contrast agents, for example dysprosium. This results in a decrease of the T2 signal [113]. Despite the advantages of contrast agents, some of the adverse effects include physiologic and mild allergic-like reactions, high toxicity levels upon accumulation in cells, and their biological stability [114]. In order to overcome some of the contrast agents’ adverse effects, metal nanoparticles have been employed as an alternative method [23]. Manipulation and detection by remote magnetic fields, high biocompatibility, and excellent magnetic properties are among some of the MNPs properties [35]. In addition, the possibility of conjugating antibodies, peptides, polysaccharides, and folic acid to MNPs allows for the targeted delivery of MNPs to their site of action, which aids in the accurate diagnosis of disease [115]. The magnetic nanoparticle most commonly used as a contrast agent is iron oxide [116].

Superparamagnetic iron oxide nanoparticles (SPIONs) and ultrasmall superparamagnetic iron oxide nanoparticles (USPIONs) are two different types of iron oxides that can increase the sensitivity of MRI. One of the unique properties of SPIONs is its fast alteration of proton’s orientation, following its exposure to an external magnetic field. This gives SPIONs a high relaxivity value and consequently increases the sensitivity of MRI [117]. SPIONs act as contrast agents by reducing the intensity of the T2 signals in the tissue. The size, shape, surface modification, magnetic susceptibility, and size distribution of SPIONs are some of the main factors influencing their functionality as contrast agents [118]. Different types of SPIONs and USPIONs have previously been approved to be used. Some of the most common examples that have used in the past include Feridex I.V., Resovist, Sinerem, and Clariscan [111]. Following the administration of MNPs into the body, the particles are engulfed by macrophages via the reticuloendothelial system. The nanoparticle’s size is an important factor that influences its recognition by the body’s immune system. USPIONs consist of iron oxide cores functionalized by different coating agents like carbohydrates or polymers. The functional group prevents the particle’s aggregation and provides a platform for the attachment of drugs and different ligands for targeted delivery. The advantage of USPIONs, in comparison to SPIONs, is their ability to stay within the blood circulation for longer due to their smaller size. USPION’S small size prevents them from becoming readily recognized by the immune system, hence their longer half-life. According to research, the USPION’s half-life is 36 h, whereas the half-life of SPIONs is only 2 h. The small size of USIPONs also allow these particles to easily diffuse into organs and accumulate within inflamed tissues. The great concentration of USPIONs in the tissues will shorten the T2 relaxation time and improve the MRI quality of imaging [119]. Novel imaging modalities, based on MNPs, have been in development in recent years. For example, researchers introduced a new imaging modality referred to as magnetic particle imaging in 2005. In addition, a 2D spatial resolution of 1 mm with the ability of imaging MNPs as far as 40 μmol (Fe) L^−1^ was reported. Despite those major advancements in the field, the attainment of early diagnosis in cancer treatment is crucial but has yet to be achieved. This remains a challenge due to the sensitivity of diagnostic tools, and the symptoms generally becoming apparent once the cancer is well advanced [43].

### 2.18. Clinical Trials of Magnetic Nanoparticles

Early clinical results have shown that nanoparticle therapies can demonstrate increased efficacy in comparison to conventional therapies and can substantially decrease negative adverse effects, due to their targeted approach [33]. This section will detail the latest advancements of MNPs in clinical trials to highlight the various openings for innovative research on MNPs in biomedical applications [22] (Table 3).

Nanoparticles and drugs must undergo a comprehensive and meticulous trial in order to obtain approval from the food and drug administrator (FDA). After the therapeutic agent has been developed in the lab, it must be tested on animals before proceeding to a clinical trial. Once the therapeutic agent has passed the animal study, it must be tested in 4 phases during a clinical trial. Phase 1 involves the assessment of metabolic activity, adverse effects, and excretion from the body in 20–80 healthy volunteers whilst the drug is administered. Phase 2 tests the therapeutic agent against a placebo in 100 patients to highlight the efficacy of the drug. Phase 3 involves 1000 patients to allow for a thorough investigation of the therapeutic agent’s efficacy. Phase 4 is the post-marketing monitoring of the drug, which entails periodic updates to the FDA about the therapeutic agent’s efficacy and adverse effects (Figure 6). The total guideline can be discovered in the FDA regulation of clinical trials [22].

Iron-based nanoparticles with various surface ligands were clinically approved to be utilized as T2 contrast agents in MRI and have been in clinical use for approximately 20 years [22,33]. 60 nm carboxydextran coated iron oxide nanoparticles named Ferucarbotran were approved for hepatocellular carcinoma and cell labelling with the trade name of Resovist in the USA and European Union, and Cliavist in France. 80–150 nm dextran coated iron oxide nanoparticles named Ferumoxide were approved for the imaging of mononuclear phagocyte systems and cell labelling with the trade name of Endorem in England and Feridex in the USA. Smaller dextran coated magnetic nanoparticles of 20–40 nm were approved for perfusion and lymph node imaging with the trade name of Sinerem in the European Union and Combidex in the USA [22]. Combidex has been utilized in various clinical trials for the purpose of lymph node metastases imaging, and is considered to be one of the most prevalent MNPs [43]. Although there have only been several clinical trials for the biomedical application of MNPs, the outcomes thus far have been highly encouraging [83].

**Table 3 polymers-13-04146-t003:** An outline of various clinical trials on MNPs, including their aims, methods, and outcomes.

Type of Magnetic Nanoparticles Used	Year of Study	Aim	Method	Outcome
Superparamagnetic iron oxide (SPIO) [120]	2020	To assess the efficacy of different doses of Magtrace in comparison to Tc-99 m and evaluate its non-inferiority.	Early-stage breast cancer patients were eligible. Randomised to receive three different doses of new SPIO.	The 3 doses of Magtrace demonstrated non-inferior rates, in comparison to the conventional technique.
Superparamagnetic iron oxide nanoparticles (SPIONs) [121]	2020	Evaluating the enhancement of the monitoring count on the skin surface in SLN detection using SPIONs.	62 patients were enrolled. Patients were split into 4 groups. The monitoring counts on the skin surface were measured and compared among the groups.	Moving a small neodymium magnet is effective in promoting the migration of magnetic tracers and increasing monitoring counts on the skin’s surface.
Iron oxide (ferumoxytol) nanoparticles [122]	2020	To evaluate if the ferumoxytol nanoparticles will improve the differentiation of benign and malignant lymph nodes in paediatric cancer patients.	42 children received a ^18^F-FDG PET/MRI, 2 or 24 h after intravenous injection of ferumoxytol.	The accumulation of ferumoxytol nanoparticles at the hilum can be used to diagnose a benign lymph node.
Superparamagnetic iron oxide nanoparticles (SPIONs) [123]	2019	Investigating whether SPIONs provide stronger SLN detection, in comparison to radioactive tracers.	SPIONs were detected by the newly developed handheld probe. The SLN and standard radioisotope detection rates were compared.	SPIONs are not inferior to the RI method.
Ultrasmall superparamagnetic iron oxide (USPIO) [124]	2019	To investigate macrophage-mediated inflammation as a possible biomarker of migraine.	The presence of macrophages in cerebral artery walls and in brain parenchyma of patients with migraine without aura was investigated, using USPIO-enhanced 3T MRI.	Migraine without aura is not associated with macrophage-mediated inflammation specific to the pain side of the head.
Polymeric magnetite nanoparticles (PMNPs) [125]	2018	To investigate the target coverage accuracy of delivering PMNPs encapsulating TMZ for the treatment of glioblastoma.	PMNPs were delivered to the centre of tumours in 10 pet dogs with spontaneous intracranial tumours. MRI was performed to examine PMNP distribution.	PMNP infusion did not cause any complications for 9 of the 10 dogs. The infusion accurately targeted the tumour mass for 70% of cases.
Super paramagnetic iron oxide nanoparticles (SPIONs) [126]	2018	To determine if the injection of SPIONs during the preoperative period for the localization of the sentinel node is feasible.	12 patients were injected with SPIONs to follow the decline of the magnetic signal in the sentinel node over time.	SPIONs detection, following preoperative injection, achieved a 100% success rate.
Ultrasmall superparamagnetic iron oxide nanoparticles (USPIONs) [127]	2017	To assess the feasibility and validity of macrophage imaging using USPIONs (ferumoxytol) in the cerebral aneurysmal wall.	17 patients were imaged on day 0 and 24 h after the first imaging, with an infusion of ferumoxytol.	Ferumoxytol uptake was identified in the cerebral aneurysmal wall of rats and in cultured macrophages.
Superparamagnetic iron oxide (SPIO) [128]	2016	To evaluate a new method for localization of breast cancer SLN using SPIO and Sentimag^®^.	SLN localization was performed on 115 patients using both the standard method and the magnetic technique.	The new magnetic tracer is feasible and promising as an alternative.
Ultrasmall superparamagnetic iron oxide nanoparticles (USPIONs) [129]	2016	Evaluating the off-label use of ferumoxytol as an intravenous MRI contrast agent for young adults and pediatric patients.	The heart rate and blood pressure of 86 patients were compared before and after receiving the ferumoxytol injection.	Ferumoxytol is an effective MR contrast agent.
Superparamagnetic iron oxide (SPIO) [130]	2014	Evaluating the new SentiMag^®^ technique’s potential equivalency to the gold standard.	150 patients (99 m) Tc were compared with the magnetic technique, utilizing SPIOs for the localization of SLNs.	Magnetic SLNB can be performed safely, easily, and equivalently well to the radiotracer method.
Ultrasmall superparamagnetic iron oxide (USPIO) [131]	2014	Investigating the safety and potential therapeutic effect of intravenous USPIO-based iron administration for infarct healing in STEMI patients.	In the first week and 3 months after acute MI, patients were undergoing multi-parametric CMR studies.	Intravenous USPIO based iron administration demonstrated improved infarct healing in acute STEMI patients.
Ultrasmall paramagnetic iron oxide (USPIO) [132]	2013	To investigate the diagnostic accuracy of combined USPIO MRI and DW MRI for LN staging in bladder and/or prostate cancer patients	Combined USPIO MRI and DW MRI findings from 75 patients were examined and compared to histopathologic LN findings	USPIO MRI and DW MRI combined enhances metastases detection in LNs of bladder and/or prostate cancer patients in short reading times

### 2.19. Toxicity of Magnetic Nanoparticles

The toxicity of MNPs is a source of concern with regards to future biomedical applications, considering the lack of research in this area. The toxic effects of MNPs could lead to reduced therapeutic effectiveness, and the activation of inflammatory or immune responses due to MNPs accumulating in organs. If MNPs enter cells, their toxic effects could disrupt nuclear activities, or cause the cell membranes to leak or become obstructed, which would result in adverse metabolic activity, cell proliferation, and viability outcomes (Table 4) [9]. For instance, nickel nanoparticles could trigger the apoptosis of A549 cells and HepG2 cells via oxidative stress and ultimately inhibit cells in the subG1 phase [133]. Thus, toxicology studies are necessary for all manufactured MNPs [9].

In order to conduct toxicity studies, toxicity assays are utilized. When performing toxicity assays, Trypan blue, Propidium Iodide (PI), and 3-(4,5-Dimethylthiazol-2-yl)-2,5-diphenyltetrazolium bromide (MTT) are the most widely used stains. The Lactate dehydrogenase (LDH) and 5-bromo-2′-deoxyuridine (BrdU) assays, utilized for metabolic activity, are alternative widespread tests. The purpose of toxicity assays is to examine essential cellular activities, for example cell viability and cell death. Cell proliferation is evaluated via the utilization of MTT and BrdU, whereas apoptosis or cell death is evaluated via the utilization of PI, LDH, and the tetrazolium compound MTT [29].

The toxicity of nanoparticles is dependent on various elements such as the method of administration, surface chemistry, biodegradability, etc. For example, no cytotoxic effects were detected in nanoparticle concentrations below 100 μg/mL on several cell lines, demonstrating that nanoparticle toxicity can also be concentration-dependent [134]. The nanoparticles’ risk to benefit profile must be reviewed, as in the case of any novel biomedical discovery, in order to determine whether the risks are justifiable. Generally, the most significant properties to consider in regard to cytotoxicity are the composition, shape, surface area, size, and coating of nanoparticles. Modifying the nanoparticle’s surface is essential to ensuring the toxic effects are kept to a minimum [25].

Cobalt ferrite nanoparticles are not widely utilized in biological studies, owing to their toxicological effects from the degradation of cobalt ions. Thus far, the application of cobalt ferrite nanoparticles for the treatment of cancer is only seen in a limited number of preclinical studies. However, if the etching and release of cobalt ions from the surface of MNPs can be controlled, their toxicity could be considered beneficial, as the cobalt ions could be employed as cytotoxic agents, much like chemotherapeutic agents. Additionally, the non-specific cobalt toxicity could be reduced by the application of site specific intratumoral injections. Essentially, the toxicity that certain elements exhibit could be advantageous [135].

Iron-based nanoparticles can be taken up by a broad range of cells via simple incubation. Cell types include lung cells, endothelial cells, nerve cells, kidney cells, liver cells, stem cells, fibroblasts, macrophages, as well as different cancer cell lines. In addition, a notable selection of MNPs were utilized with an assortment of cell types. The magnitude of toxicity is known to differ with MNPs and/or cell type [136]. Thus, it is vital to conduct toxicity studies to assess the toxicity of certain MNPs on specific cell types [29]. For example, the toxicology of 30 nm and 500 nm iron oxide nanoparticles were researched via incubation with the A549 alveolar epithelial cell line. The in-vitro test demonstrated a dose and size dependent toxicity. No toxicity was shown at a low dose of 40 μg mL^−1^, but a greater degree of toxicity was found at a high dose of 80 μg mL^−1^. A higher level of toxicity was also found in smaller sized nanoparticles compared to larger nanoparticles [137]. In an alternative study, it was noted that uncoated iron-based nanoparticles caused significant cell death in dermal fibroblasts, whilst lung cells appeared to be unaffected. This signifies the importance of the relationship between MNPs and cell types [29].

A series of in-vitro and in-vivo experiments are conducted to order to investigate toxicity. In-vitro toxicity tests are a cost-effective approach to collecting preliminary toxicity data in a simple, time efficient manner with few ethical issues. In order to advance to in-vivo studies, the in-vitro test results must display minimal to no toxicity. In those instances, small animal experiments were performed and monitored over a period of time to assess the long-term effects of MNPs in biological environments. To search for indications that the MNPs are spreading and accumulating in injection sites and significant metabolic sites, such as the brain, liver, kidney, and pancreas, toxicity validation tests such as histology are employed. It is possible to obtain contradictory in-vivo and in-vitro results. This could be a result of the in-vivo bodily processes working to remove foreign substances that are absent in-vitro. If the in-vivo studies present with promising outcomes, a complete assessment of the safety and therapeutic effectiveness of MNPs can be executed, and regulatory bodies such as the FDA can approve the treatment for clinical use [29].

Contrast agents based on MNPs that are on the market at present, such as Resovist^®^, Magnevist^®^, or Sinerem^®^, have complied with the current requirements relating to patient use. This is equally true for magnetic drug delivery systems that have previously been placed on the market such as TargetMAG^®^ [25].

An important issue that is frequently faced with neglect in the literature is the long-term effects of MNPs once they have achieved their role within the body. It is crucial to understand how MNPs interact and behave within the human body. Moreover, it is important to identify where the nanoparticles ultimately end up and how they are excreted [32]. Given that many industrial nanoparticles are non-biodegradable, the strong possibility of prolonged accumulation in tissues highlights the need for long-term studies since the toxic effects of some nanoparticles may only become evident following long-term exposure. This is frequently overlooked within short-term in-vitro studies and the number of long-term studies conducted to date are limited. However, it is vital to understand the medical and environmental consequences of prolonged exposure to MNPs [138].

**Table 4 polymers-13-04146-t004:** Some of the toxic effects of several MNPs and their impact on the body.

MNPs	Adverse Effects	Biological Systems Affected	Year of Study
Metal oxides [139,140]	Lung inflammation Hormonal imbalance	Reproductive system	2015, 2017
Iron oxide [141,142,143,144,145,146]	Necrosis Haemolysis Oxidative stress Denaturation DNA damage Increased manganese levels	Circulatory system Digestive system Immune system Endocrine system	2014, 2015, 2016, 2017, 2018, 2019
Cobalt oxide [147]	Necrosis	Immune system	2015
Cobalt ferrite [141]	DNA damage Unstable heartbeat Oedema	Urinary system Circulatory system	2019
Nickel [141,148,149]	Lung inflammation Cardiac toxicity Hormone imbalance	Circulatory system Reproductive system	2014, 2019
Magnetite [141,150]	Alterations in immunological pattern	Immune system	2015, 2019

## 3. Conclusions

MNPs are novel therapeutic agents that could revolutionize the diagnostic and therapeutic applications of cancer. MNPs are achieved via numerous synthetic approaches. Co-precipitation, thermal decomposition, hydrothermal, and polyol synthesis are the most prevalent techniques. Functionalization has significantly enhanced the biocompatibility of MNPs. The most widespread functionalizing agents to date are organic and inorganic polymers. The utilization of MNPs as promising tools in biomedical applications such as drug delivery, magnetic hyperthermia, and diagnosis is largely attributed to functionalization. In the past decade, numerous clinical trials have been carried out for MNPs. This is a great testament of the potential of MNPs in the scientific field. However, long-term toxicology studies have yet to be achieved and t the eco-toxic effects of MNPs have often been overlooked. Therefore, it is crucial that these issues are addressed.

The treatment of cancer using nanoparticles is a multibillion dollar industry, with a few studies moving to clinical trials within the next 5 years.

## Figures and Tables

**Figure 1 polymers-13-04146-f001:**
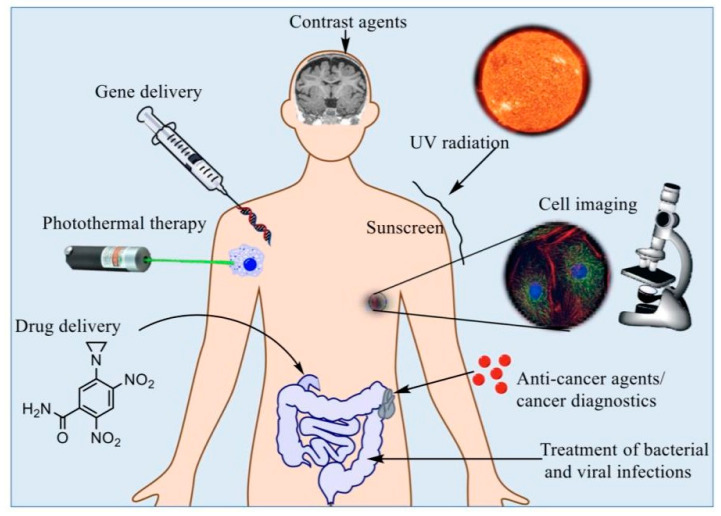
Several biomedical applications for nanoparticles at present [23].

**Figure 2 polymers-13-04146-f002:**
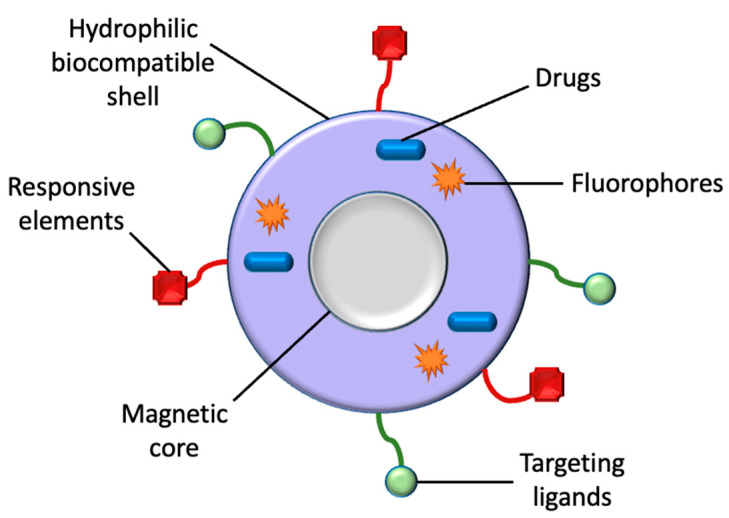
A classic illustration of a magnetic nanoparticle’s structure. Aside from drugs, MNPs can also be utilized for the transportation of targeting ligands, fluorophores, and responsive elements for their respective biomedical applications.

**Figure 3 polymers-13-04146-f003:**
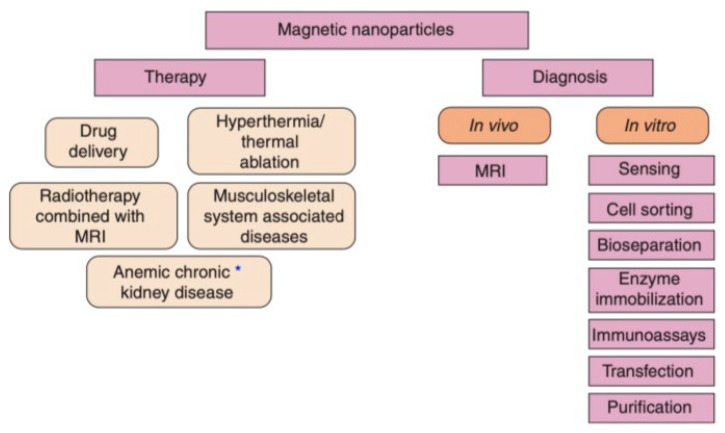
The range of available biomedical applications for MNPs. The asterisk (*) represents either diagnostic or therapeutic applications that are undergoing clinical trials [25].

**Figure 4 polymers-13-04146-f004:**
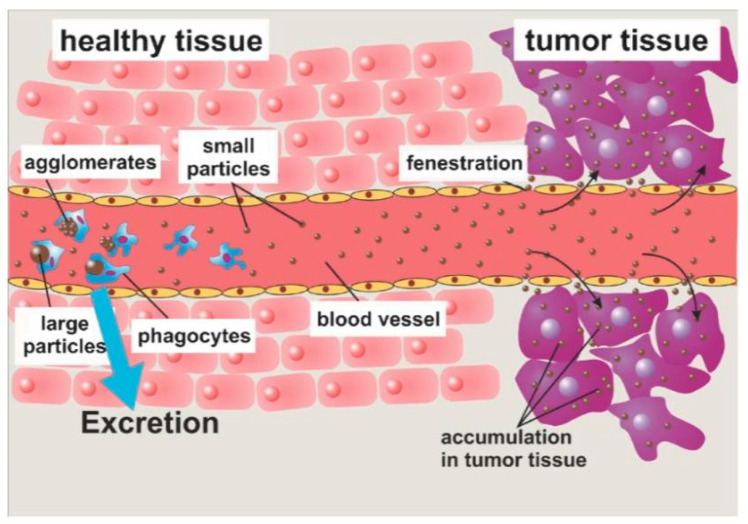
Passive targeting occurs as a result of the enhanced permeability and retention (EPR) effect. Large particles are engulfed by phagocytes, whilst small particles proceed to the target. The EPR effect allows the MNPs to accumulate within the target tumour [33].

**Figure 5 polymers-13-04146-f005:**
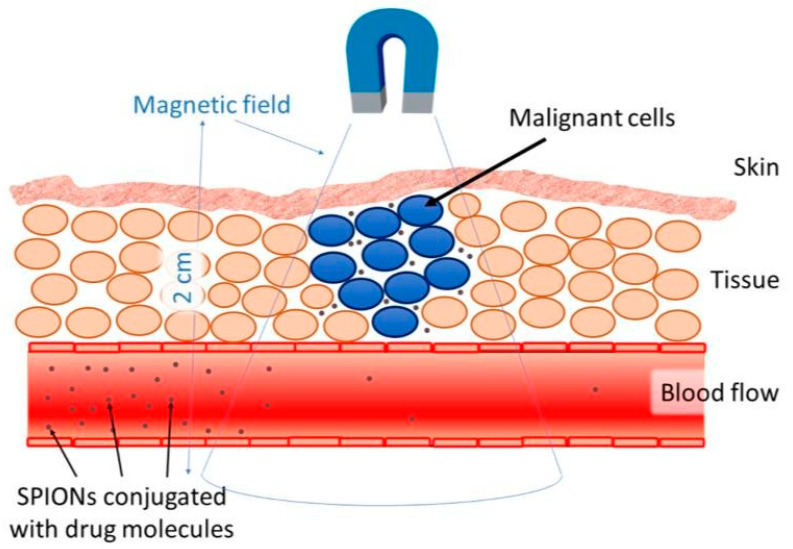
SPIONs immobilizing drugs with the utilization of an external magnetic field [30].

**Figure 6 polymers-13-04146-f006:**
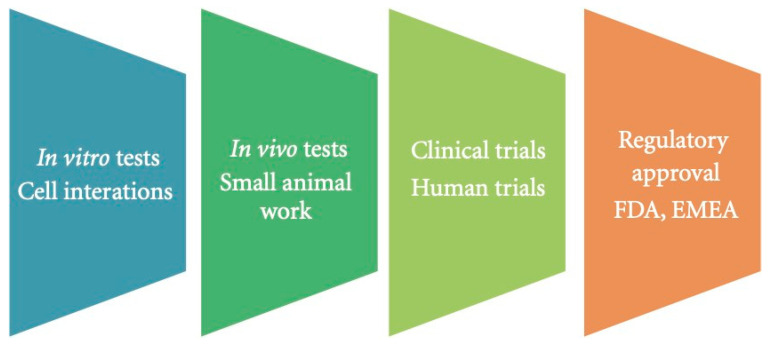
A representation of the clinical trial process [29].

## Data Availability

The data presented in this study are available on request from the corresponding author.

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
