# Peer review of "Emerging Application of Magnetic Nanoparticles for Diagnosis and Treatment of Cancer"

_polymers, 2021, doi:10.3390/polym13234146_

Round 1

Reviewer 1 Report

The authors propose in this paper a review of some biomedical applications of magnetic nanoparticles. The topic is very actual and interesting, but at the same time it is very complex and an exhaustive presentation would be very difficult.

Although the authors chose to present only some specific applications (namely: hyperthermia (HT), drug delivery, and diagnostic (MRI)) they didn't succeed to structure it in such a manner to captivate the attention of a potential reader, either specialized in this field or less familiarized with it. The paper seems to be more like a collection of paragraphs from other reviews, without a clear structure, with statements that sometimes are not rigorously correct, reducing thus its scientific soundness. In many cases, the references are not the most relevant. 

For these reasons, I consider that the paper is not suitable for publication. Please find below some main observations:

-The first 2/3 of the abstract looks like an introduction. The authors should present in the abstract a short background and summarize the main content of the article and the conclusions or interpretations.

- the References chapter contains 95 titles while within the paper the references are numbered up to 110.

- several references do not match the context in which are mentioned (e.g. in Table 4. Some of the toxic effects of several MNPs and their impact on the body in the last two rows are referred the titanium oxide and zirconia oxide which are not magnetic; Table 3 calcium chloride(?), silver nanoparticles, silica gold nanoparticles are not magnetic nanoparticles, moreover, for all these rows within the table the references indicated do not match the content of the Table).

-in the magnetic hyperthermia chapter or the clinical trial chapter there is no indication about the clinical trials performed by MagForce for the treatment of glioblastoma multiforme or prostate cancer, the most relevant in this field;

-the MRI chapter is only a short presentation of the role of contrast agents in enhancing the contrast in T1 or T2 weighted images, without presenting the main advantages or limitations in the use of magnetic nanoparticles in this field;

-in the drug delivery chapter there is basically one single example

- examples of unclear, potentially misleading sentences:

"The magnetic field is only capable of penetrating the body up to 2 cm from the skin"

"Figure 5. MNPs, such as superparamagnetic iron oxide nanoparticles (SPIONs), can immobilize drugs with the utilization of an external magnetic field"

Reviewer 2 Report

Comments: The purpose of this review was to address the fabrication and functionalization of  magnetic nanoparticles for diagnostic and treatment of cancer. The authors try to highlight those nanoparticles that have demonstrated more promising results and also the one that are already into clinical trials.

  • In my opinion, this review it is very promising. However, there are few issues should be solved. The references listed in this review are very scarce. In the last 2 decades there is been a “huge”amount of literature related on the development of magnetic nanoparticles for cancer therapies purposes. References should have been updated.
  • The references cited in the manuscript are not in the right format of the journal. They should appear in square brackets [.]. Also, the format of the references listed are not also in right format. Authors should follow the references format of the Polymer /MDPI journal “Author 1, A.B.; Author 2, C.D. Title of the article. Abbreviated Journal Name Year, Volume, page range”.
  • Another week point of this review, there were not corresponding relation between references in table 3 and their citations in references list.
  • Authors did not explain why did they cite “Calcium chloride (CaCl2) in topic of magnetic nanoparticles
  • New subdivision termed “Fluoro-magentic Nanoparticles” should have been added for cancer bio image and New Ref should be cited “: 1039/C6RA06345D”

Round 2

Reviewer 1 Report

The resubmitted manuscript is improved as compared to its first version. However, in the reviewer’s opinion, the paper cannot be published in this form.

As I mentioned in the first review, the topic the authors try to address is very generous with an exponential development in the last decade,  and it is really a very hard task to make an exhaustive presentation. On the other hand treating: MNP synthesis, their functionalization, and three main theranostic applications (magnetic hyperthermia, drug delivery, and MRI contrast agents) in  25 pages seem to be an almost impossible mission.

Regarding the human clinical applications of magnetic hyperthermia suggested in my first review, the authors didn’t address correctly this topic. In fact, a very recent review (Cancers 2021, 13, 4583. https://doi.org/10.3390/cancers13184583) made a very good presentation and an update of the progress obtained in this field.  

Nevertheless, some chapters are well structured and could trigger the interest of the journal readers like the clinical and preclinical trials (related mainly to the MNPs use as contrast agents in MRI, the updated Table 3) and the toxicology issues (Table 4). Therefore I suggest the authors restructure their manuscript and keep eventually the synthesis and the functionalization chapters, only to shortly mention the drug delivery and local magnetic hyperthermia applications and to elaborate a little bit more on their use as MRI contrast agents and the toxicological issues for which they correctly updated the references.

Though the authors improved the chapter with the role of  MNPs in increasing the contrast in MRI this chapter still needs to be improved. While indeed the authors added additional information about T1 and T2 contrast agents, they introduced the relaxivities, the chapter doesn’t provide enough information for a  reader not familiar with the field. The relaxivities are not defined. Some statements are not scientifically rigorous (paramagnetic nanoparticles reduce T1). In fact, paramagnetic ions like Gd3+, Mn 2+ and Dy 3+ or their complexes, with very large magnetic momentum, are used to reduce T1 as bright contrast agents due to their interaction with the protons in the first hydration layer, and paramagnetic nanoparticles are used as dark contrast agents in T2 weighted images, their large magnetic momentum influencing the protons situated farther from their magnetic core due to their coatings, thus not being able to strongly influence T1.  The authors should also emphasize the recent progress in this field by using ultrasmall SPIONs.

Reviewer 2 Report

Authors have followed the correction step by step. The manuscript is more acceptable 

Author Response

Thank you. The manuscript has significantly improved.